# Construction of the core competencies training system for thoracic surgery specialist nurses: A mixed-methods study

Yingjin Li[1], Yan Chen[2]*, Qiuju Chen[3], Di Zhu[4], Ping Xia[5], Yuqin Mao[5], Liumei Yin[5], Saijun Ba[6]

1 School of Nursing, Nanjing University of Chinese Medicine, Nanjing, China, 2 Department of Nursing, Nanjing Drum Tower Hospital Clinical College of Nanjing University of Chinese Medicine, Nanjing, China, 3 Jiangbei Department of Nursing, Nanjing Drum Tower Hospital, Affiliated Hospital of Medical School, Nanjing University, Nanjing, China, 4 Department of Rheumatology and Immunology, Nanjing Drum Tower Hospital, Affiliated Hospital of Medical School, Nanjing University, Nanjing, China, 5 Department of Thoracic Surgery, Nanjing Drum Tower Hospital, Affiliated Hospital of Medical School, Nanjing University, Nanjing, China, 6 Department of Surgical Day Ward, Nanjing Drum Tower Hospital, Affiliated Hospital of Medical School, Nanjing University, Nanjing, China

* njchenyan@126.com

## Abstract

### Objective

To develop a training system for cultivating the core competencies of thoracic surgery specialist nurses.

### Methods

A mixed-methods study was employed, comprising two stages: (1) A literature review and semi-structured interviews with thoracic surgical healthcare professionals were conducted to develop a preliminary core competencies training system for thoracic surgery specialist nurses; (2) A two-round Delphi expert consultation was conducted to determine the final training system.

### Results

Consensus was reached on the core competency framework (training objectives) (6 first-level, 17 second-level indicators), 92 items of curriculum content along with their corresponding teaching methods, 17 aspects of organizational management, and 8 evaluation methods. The response rate was 100%. The authority coefficient was 0.895. The Kendall's coefficients for the two rounds of expert inquiry were 0.141 and 0.210, respectively.

### Conclusion

This study developed a scientific, comprehensive specialty training system for thoracic surgery specialist nurses, meeting practical nursing needs, and enriching the

**Data availability statement:** All relevant data are within the manuscript and its Supporting Information files.

**Funding:** The author(s) received no specific funding for this work.

**Competing interests:** The authors have declared that no competing interests exist.

field's evidence base. It delivers substantial clinical and practical value to nursing administrators, practitioners, patients, and the healthcare system. Future research should validate its practical value through larger-scale empirical studies, continuously gather feedback on training outcomes, refine the training system, and further enhance its adaptability and influence.

## Introduction

Thoracic Surgery is a specialized branch of medicine focused on the surgical management of intrathoracic pathologies. It principally addresses neoplastic and functional disorders of the esophagus, pulmonary parenchyma, and mediastinum. Global cancer statistics show that approximately 2.48 million new lung cancer cases were reported in 2022. This accounts for 12.4% of all new malignancies and ranks first worldwide. Esophageal cancer also remained a major health concern, with about 0.51 million new cases reported during the same period [1]. Despite advancements in minimally invasive techniques and the implementation of enhanced recovery after surgery (ERAS) protocols, thoracic procedures remain classified as high-risk Grade IV surgeries due to their proximity to vital structures such as the heart, lungs, and esophagus, coupled with complex intraoperative dynamics and significant resource utilization [2]. Studies have shown that the incidence of postoperative complications following thoracic surgery reaches up to 29.4% [3]. Among these, postoperative pulmonary complications are the most common, occurring in 4% to 55% of pulmonary surgeries and 9.8% to 52% of esophageal surgeries [4,5]. These complications increase patient morbidity, extend hospital stays, raise healthcare costs, and may delay adjuvant therapy, ultimately compromising long-term survival.

A specialist nurse refers to a registered nurse who possesses solid theoretical knowledge of nursing and proficient clinical operational skills in specific nursing specialties, has completed and passed the assessments of specialized nurse systematic training programs, and is capable of conducting high-level nursing practice. They should hold a bachelor's degree or higher [6]. Within this high-stakes clinical context, thoracic surgery specialist nurses play a crucial role in coordinating postoperative care, delivering targeted health education, providing psychological support, and ensuring consistent nursing quality [7]. Evidence has shown that the cultivation and utilization of specialist nurses in thoracic surgery can improve the overall quality of nursing care, reduce patient mortality and readmission rates, and lower medical costs and patients' economic burdens, which is highly important for the healthy development of the national economy and the high-quality development of nursing services [8].

Thoracic surgery nursing is a highly professional and technical field, involving multiple key areas, including perioperative management, respiratory function training, complication prevention, and intensive care. As important members of the thoracic surgery medical team, the core competencies of nurses directly affect the treatment effect and prognosis of patients. ICN first defined the core competencies of

specialist nurses as the knowledge, skills, judgment, and personal qualities required by nursing staff to provide safe and ethical nursing services to patients [9]. Recognizing this, China's National Nursing Career Development Plan (2021–2025) explicitly mandates the strengthening of disciplinary construction, with a priority on targeted training of specialist nurses in high-demand fields that are witnessing robust development, thereby aligning with the evolving clinical needs [10]. Thoracic surgery specialist nurses can guide and oversee registered nurses in clinical thoracic surgical disease nursing, utilizing their professional knowledge and skills to enhance communication with postoperative patients, families, and interdisciplinary teams. They also offer systematic and standardized training in postoperative monitoring, complication management, and mentoring [7]. However, at present, the cultivation of core competencies for thoracic surgery specialist nurses in China has not yet formed a unified, standardized, and scientific curriculum training system. There are significant differences in training content and formats among various hospitals and training institutions, making it challenging to meet the needs of clinical practice.

The Tyler Model [11], which emphasizes educational objectives as the foundation of curriculum design and evaluation, integrates training content, methods, and assessment around these objectives. It has been widely applied in the development of training frameworks [12,13]. Guided by this model, the study aims to develop a comprehensive, systematic, scientific, and standardized training system for thoracic surgery specialist nurses, based on the Tyler Model. This core competencies training system can provide a scientific foundation for the training and management of thoracic surgery specialist nurses, thereby enhancing their clinical practice and comprehensive nursing competence, and further promoting the development and advancement of the thoracic surgery discipline.

## Methods

The study employed a mixed-methods approach, comprising two stages: (1) A literature review and semi-structured interviews with thoracic surgical healthcare professionals ($N=21$) were used to construct a preliminary core competencies training system. (2) A consensus regarding the core competency of thoracic surgery specialist nurses as well as the corresponding training system was achieved by the Delphi method ($N=19$) (Fig 1). The study was approved by the Ethics

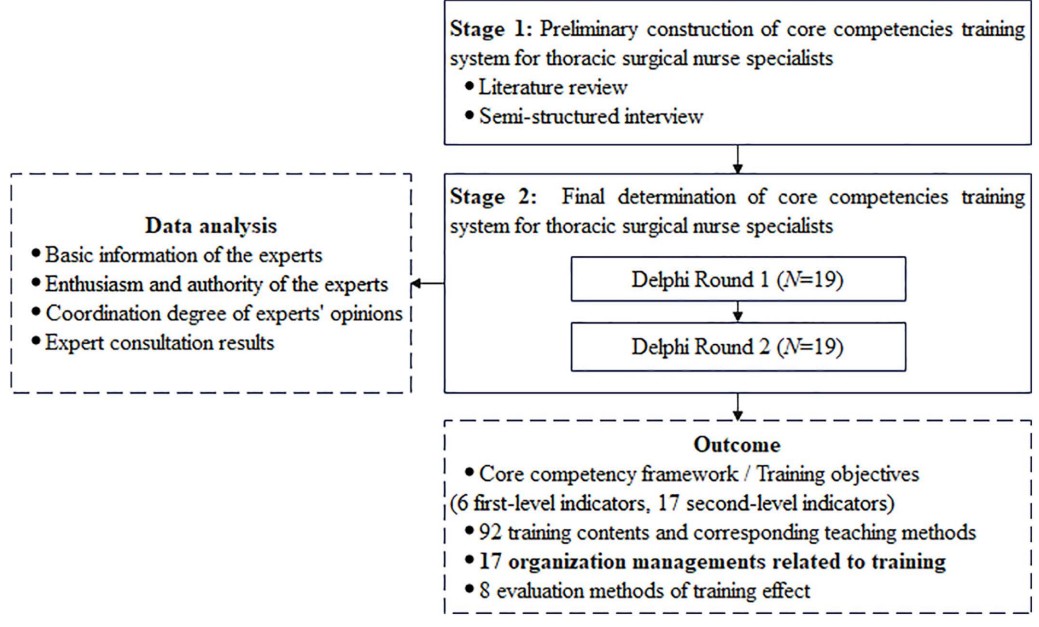

**Fig 1. Flowchart of the study design.**

Committee of Nanjing Drum Tower Hospital Affiliated to Medical College of Nanjing University (Ethics No. 2024-720-01) and conducted in accordance with the Declaration of Helsinki.

## Establishing a research group

A multidisciplinary research team was established, comprising 8 members, including 4 specialist nurses. In terms of professional titles, there is 1 senior professional title, 3 deputy senior professional titles, 2 intermediate professional titles, and 2 junior professional titles. Regarding educational backgrounds, 5 members hold bachelor's degrees, 2 hold master's degrees, and 1 is a master's candidate. The team leader oversaw the overall study, and other members were involved in the literature review and organization, qualitative interviews, drafting of the outline of the initial article, process of selecting and contacting experts for inquiries, preparation and distribution of the expert questionnaires, summary and analysis of the expert opinions, and provision of timely feedback.

## Literature review

The literature analysis was conducted from October 1 to December 31, 2024. The research team first identified the research questions and objectives, and then conducted a literature search. This involved screening the literature that met the research topics and criteria, paying attention to the publication date, source, and quality of the literature. Databases such as PubMed, Embase, Cochrane, CINAHL, Web of Science, CNKI, Wanfang Database, VIP, and CMB were searched, with the retrieval period spanning from the establishment of each database to July 2024. The main retrieval terms included "Thoracic Surgery", "Specialist Nurse", "Advanced Practice Nurse", "Core Competencies", and "Curriculum". In the afore-mentioned databases, relevant studies were comprehensively retrieved through multi-dimensional screening, including titles, abstracts, keywords, subject terms, and reference lists (S1 File). Mixed methods research was quality-assessed using the Mixed Methods Appraisal Tool (MMAT) [14], and other studies via the JBI Evidence-based Healthcare Center critical appraisal tools [15]. Both were conducted independently by two evidence-based nursing-trained researchers. Any discrepancies or disputes during assessment were resolved by consulting third-party experts (S2 File in S1 File).

## Semi-structured interviews

From January 1 to March 31, 2025, the research team conducted face-to-face semi-structured interviews with thoracic surgery nurses and medical/nursing experts using purposive and snowball sampling methods. Sampling was terminated when information saturation was achieved, with no new information emerging from additional samples, resulting in a total of 12 thoracic surgery nurses and 9 thoracic surgery medical/nursing experts interviewed. All interviewers under-went training related to semi-structured interviews, gaining a thorough understanding of the survey content, instruments, and interview techniques. They clarified data collection and organization methods and conducted simulations before the formal interviews. The interviews aimed to gather the perceptions and experiences of nurses and medical/nursing experts regarding the competency requirements of thoracic surgery specialist nurses. The inclusion criteria and the corresponding interview questions are shown in Table 1. Before the interviews, the participants were fully informed about the research, and their informed consent was obtained (S3 File in S1 File). Once written consent was given, the recording of the inter-views officially began. The duration of each interview varied from 30 to 60 minutes. Detailed notes were taken during each session, and a verbatim transcription was completed within 24 hours of the session. The transcripts were returned to the participants for comments or corrections.

## Delphi method design

**Experts' qualifications.** The number of Delphi consultants should be controlled at 15–30 to avoid the homogeneity of research objects [16]. From April 1 to June 30, 2025, the group distributed questionnaires to thoracic surgery nursing

**Table 1. The inclusion criteria and the corresponding interview questions of the semi-structured interviews.**

| Interview subject | The inclusion criteria | The corresponding interview questions |
|---|---|---|
| Thoracic surgery nurses | (1) Holding a valid nursing licence;<br>(2) At least 3 years of thoracic surgery nursing work experience;<br>(3) Voluntary participation. | (1) What specialized nursing services do you think are necessary for thoracic surgery patients?<br>(2) What challenges have you encountered when providing specialized nursing care to thoracic surgery patients, and how did you address these challenges?<br>(3) What core competencies or qualities do you believe are essential for becoming a thoracic surgery specialist nurse?<br>(4) In your opinion, which areas of competency in thoracic surgery nursing currently require the most improvement?<br>(5) Is there any additional feedback or comments you would like to provide based on of the above discussion? |
| Thoracic surgery medical experts | (1) Intermediate or senior professional title;<br>(2) A graduate degree or higher;<br>(3) At least 10 years of thoracic surgical work experience;<br>(4) Voluntary participation. | (1) In your view, what specific areas of specialized nursing care are essential for thoracic surgery patients?<br>(2) From your perspective, which aspects of nursing care do thoracic surgery nurses currently perform well, and which areas require further improvement?<br>(3) What core competencies or qualities do you believe a thoracic surgery specialist nurse must possess to meet clinical demands? |
| Thoracic surgery nursing experts | (1) Intermediate or senior professional title;<br>(2) A bachelor's degree or higher;<br>(3) At least 10 years of thoracic surgery work experience;<br>(4) Voluntary participation. | (4) In thoracic surgery nursing, which competency domains do you think currently need the most development or enhancement?<br>(5) If you were responsible for training thoracic surgery specialist nurses, what key components or strategies would you include in the training program?<br>(6) Is there any additional insight or feedback you would like to share based on the topics discussed above? |

and medical experts for expert consultation. The inclusion criteria were as follows: (1) A bachelor's degree or higher; (2) Associate senior or higher professional title; (3) At least 10 years of work experience in nursing or medical practice; (4) Engaged in clinical nursing/medicine, nursing education, nursing management, or research in thoracic surgery; and (5) Willing to participate and demonstrate strong enthusiasm for the research.

**Construction of the expert consultation questionnaire.** The expert consultation questionnaire was composed of three parts: (1) Basic information of the experts, which included name, age, professional title, years of experience in thoracic surgery, etc.; (2) The preliminary core competencies training system for thoracic surgery specialist nurses which consisted of a core competency framework (training objectives), curriculum content along with their corresponding teaching methods, organizational management, and evaluation methods. The experts were asked to rate the importance of these items on a 5-point Likert scale ranging from "extremely unimportant" to "extremely important", and a column for suggestions and supplements was provided; (3) Expert familiarity with the content of the survey and judgment (S4 File in S1 File).

**Delphi consultation process and feedback.** The questionnaire was distributed via email or WeChat, and the consultation was concluded when experts' opinions reached substantial consistency. This research involved two rounds of expert consultation. After collecting the first-round questionnaires, the research team promptly summarized, analyzed, discussed, and revised the feedback to develop the second-round questionnaire (S5 File in S1 File). During the second round, the results of the previous consultation (including discussions and analyses) were provided to experts as feedback for their reference and evaluation. In each round of consultation, indicators were evaluated based on the following criteria: those with an average importance score ≥3.5, a full-score percentage ≥20%, and a coefficient of variation <0.25 were retained; those failing to meet these criteria were adjusted, removed, or merged based on expert feedback and group discussions. Since experts' opinions in the second round showed high consistency, the consultation was concluded. To ensure the quality of the questionnaires, experts were given two weeks to complete each round of the Delphi survey, with a one-month interval between the two rounds. Based on the feedback from both rounds, the core competencies training system for thoracic surgery specialist nurses was revised and finalized.

## Data analysis

The interview data were analyzed using the seven-step method developed by Colaizzi [17]. This approach was used to transform the recordings into documents immediately, which were compiled after manual checking, discussed repeatedly, classified, and refined. Ultimately, 6 themes were obtained, which could serve as a reference for the development of training courses.

The Delphi data were analysed via SPSS version 27.0 software. The measurement information is presented as the mean and standard deviation, while the count information is expressed as frequency and percentage. The questionnaire return rate and the opinion submission rate reflect the experts' enthusiasm. The arithmetic mean of the coefficient of judgment (Ca) and the coefficient of familiarity (Cs) indicates the coefficient of expert authority (Cr), i.e., Cr=(Ca+Cs)/2. Cr>0.7 indicates a high degree of expert authority. The coefficient of variation (CV) and Kendall's coefficient of agreement indicate the degree of harmonization of expert opinions. Kendall's coefficient ranges from 0 to 1, with values closer to 1 indicating greater consistency in expert opinions and stronger reliability of the results. The CV is the ratio of the standard deviation to the mean importance score, reflecting the degree of fluctuation in experts' opinions on the indicators. A smaller coefficient of variation indicates a higher level of coordination among experts [18,19]. All statistical tests were conducted at a significance level of $\alpha = 0.05$, with $P < 0.05$ defined as statistically significant.

## Results

### Literature review

A total of 6,354 studies were retrieved initially. Following duplicate removal, 4,871 unique studies were retained. Through rigorous screening of titles, abstracts, and full texts, 14 studies were finally included in this research [20–33]. The flow chart of the study selection process is shown in Fig 2.

### Basic information of the experts

The 19 experts in the two rounds were the same, recruited from hospitals and universities in Beijing, Shanghai, Jiangsu, Guangdong, and Sichuan. Their ages ranged from 37 to 69 years, with a mean age of 44.05 (SD = 6.00) years. The expert team consisted of 12 thoracic surgery nursing experts, 2 thoracic surgery medical experts, 4 nursing management

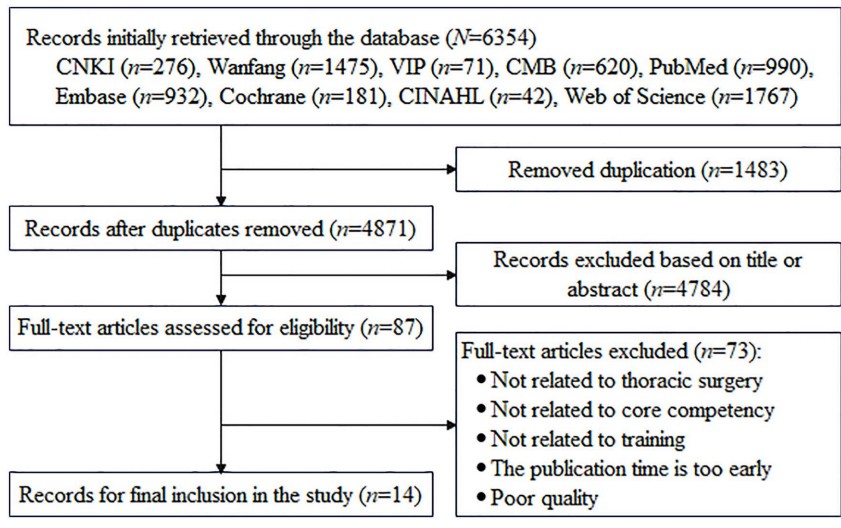

**Fig 2. Flowchart of study selection.**

experts, and 1 nursing education expert. Their work experience ranged from 12 to 40 years, with a mean of 21.84 (SD = 8.09) years. All experts had extensive clinical and teaching experience in thoracic surgery. Detailed information on the experts is provided in Table 2.

### Enthusiasm and authority of the experts

During the two rounds of consultation, 19 questionnaires were distributed, and all 19 were returned, resulting in a 100% response rate. In the first round of correspondence inquiry, 10 experts (52.63%) provided suggestions, while in the second round, 8 experts (42.11%) submitted suggestions. The Ca and Cs were 0.974 and 0.816, respectively, yielding a Cr of 0.895. These values indicate active and authoritative expert participation in the consultation, confirming the process's reliability (S6 File in S1 File).

### Coordination degree of experts' opinions

After two rounds of expert consultation, the importance score of all indicators exceeded 3.5, and the CVs of all indicators were less than 0.25 (S7 File in S1 File). Additionally, Kendall's concordance coefficient was 0.141 for the first round and 0.210 for the second round. The values in the second round were consistently higher than those in the first round for almost all indicators, suggesting an increasing convergence of expert opinions. The statistical significance of these coefficients ($P < 0.05$) for all indicators further confirms that the achieved consensus was not a random result but a reflection of genuine agreement among experts (Table 3).

### Expert consultation results

In the first round of questionnaire consultation, 10 experts provided revisions for the training system. After the group discussion, we modified the training system by revising 3 items of the core competency framework (training objectives), deleting 2 items, revising 19 curriculum contents, adding 6 new curriculum contents, deleting 6 existing curriculum contents, revising 2 organizational management aspects, and revising 2 evaluation methods. In the revision of the core

Table 2. Demographic characteristics of the experts (*N* = 19).

| Variables | Classification | Frequency (*N*) | Percentage (%) |
|---|---|---|---|
| Gender | Female | 17 | 89.5 |
| | Male | 2 | 10.5 |
| Age | 31 ~ 40 | 6 | 31.6 |
| | 41 ~ 50 | 11 | 57.9 |
| | > 50 | 2 | 10.5 |
| Work experience (years) | 10 ~ 20 | 11 | 57.9 |
| | 21 ~ 30 | 5 | 26.3 |
| | > 30 | 3 | 15.8 |
| Work Field | Thoracic surgery nursing expert | 12 | 63.1 |
| | Thoracic surgery medical expert | 2 | 10.5 |
| | Nursing management expert | 4 | 21.1 |
| | Nursing education expert | 1 | 5.3 |
| Education level | Bachelor's degree | 9 | 47.4 |
| | Master's degree | 7 | 36.8 |
| | Doctoral degree | 3 | 15.8 |
| Professional title | Deputy senior professional title | 14 | 73.7 |
| | Senior professional title | 5 | 26.3 |

**Table 3. Experts' coordination degree.**

| Round | Index | Number | Kendall's coefficient | $\chi^2$ | $P$ |
|---|---|---|---|---|---|
| First round | Core competency framework (Training objectives) | 25 | 0.204 | 92.876 | <0.001 |
| | Curriculum content | 93 | 0.141 | 246.507 | <0.001 |
| | Organizational management | 17 | 0.104 | 31.532 | 0.011 |
| | Evaluation methods | 8 | 0.253 | 33.690 | <0.001 |
| | Total | 143 | 0.141 | 345.730 | <0.001 |
| Second round | Core competency framework (Training objectives) | 23 | 0.251 | 104.777 | <0.001 |
| | Curriculum content | 92 | 0.218 | 377.772 | <0.001 |
| | Organizational management | 17 | 0.166 | 50.399 | <0.001 |
| | Evaluation methods | 8 | 0.180 | 24.000 | 0.001 |
| | Total | 140 | 0.210 | 530.634 | <0.001 |

competency framework (training objectives), the first-level indicator IV "Scientific research and innovation competency" was renamed "Innovative practice competency". The second-level indicator III-2 "Collaboration competency" was adjusted to "Coordination competency". In contrast, IV-2 "Basic scientific research competency" was revised as "Research and innovation competency", and VI-2 "Individual traits" was updated to "Professional literacy". Additionally, the second-level indicators I-6 "Application competency of related professional knowledge" and IV-1 "Self-directed learning competency" were deleted. Concurrently, corresponding adjustments were made to the curriculum content: after removing the above-mentioned deleted indicators, their associated curricular content was integrated into other second-level indicators. New content was added, including I-1–4 "Specialized pharmacological knowledge and administration standards in thoracic surgery", I-1–5 "Specialized knowledge of thoracic surgical examinations and result interpretation", I-2–16 "Development of electromagnetic navigation bronchoscopy technology and nursing implications", and II-2–4 "Curriculum design for thoracic surgery nursing". Meanwhile, indicators I-3–3 "Chest X-ray interpretation" and I-3–4 "Lung auscultation" were removed. The content of indicators I-2-10, I-2-12, and I-2-13 was respectively revised to "Common surgical procedures for thoracic diseases and postoperative nursing key points", "Development of preoperative auxiliary localization techniques for pulmonary subsolid nodules and related nursing", and "Advances in interventional therapy for thoracic tumor patients and nursing practice". In the organizational management section, based on expert recommendations and integrating the admission criteria for trainees from the Chinese Nursing Association and provincial nursing societies, the research team adjusted the trainee admission standard to "At least 3 years of nursing experience in thoracic surgery". The faculty selection criteria were revised to "At least 5 years of teaching experience in academic institutions or clinical training settings". With respect to the evaluation methods, the indicator "English literature presentation" was renamed "Chinese/English professional literature reading presentation". Considering the difficulty in assessing "Health education and science popularization", the experts modified this indicator to "Health education popularization video/manual development".

In the second round of questionnaire consultation, 8 experts provided revisions for the training system, including 3 curriculum contents, 7 teaching methods, and 1 organizational management. Revisions were limited to minor wording adjustments for the training system. For example, I-3–1 "Common assessment tools in thoracic surgery" was revised to "Commonly used perioperative assessment tools in thoracic surgery". Teaching methodologies were further supplemented and refined. The teaching methods for indicator I-1–5 "Specialized knowledge of thoracic surgical examinations and result interpretation" were adjusted to a lecture-based theoretical teaching and case-driven discussion framework. In contrast, those for indicator I-4–9 "Emergency management of thoracic drainage tube dislodgement" were modified to include lecture-based theoretical teaching and scenario simulation pedagogy. Notably, as the teaching faculty would not be limited to thoracic surgery specialists, the faculty selection criterion "At least 10 years of experience in thoracic surgery-related specialties" was revised to "At least 10 years of specialized experience in related specialties" based on

expert recommendations. For example, statistics instructors are required to have at least 10 years of experience in the field of statistics. Finally, a core competency training system for thoracic surgery specialist nurses was developed, including a core competency framework (training objectives) comprising 6 first-level and 17 second-level indicators, 92 curriculum content items and their corresponding teaching methods, 17 aspects of organizational management, and 8 evaluation methods, as shown in Tables 4 and 5.

## Discussion

### Practical significance of building the core competencies training system

The 30th ICN Congress in 2025, themed "Nursing power to change the world", highlights nurses' pivotal influence in shaping the evolving healthcare landscape and highlights the contemporary value of the profession [34]. China has advanced nursing professionalization through the targeted implementation of national policies. The *Healthy China 2030 Plan Outline* calls for strengthening nursing talent development to enhance health service capacity and promote specialization and subspecialization by 2030 [35]. The widespread adoption of the ERAS concept has increased thoracic surgery volume and patient turnover efficiency [36]. To meet growing clinical demands and address the complexity of diseases, cardiothoracic surgery has progressively evolved into independent thoracic and cardiac surgery departments [8]. This specialization creates both opportunities and challenges for thoracic surgery specialist nurses. In Western countries, thoracic surgery specialist nurse training is well established, focusing on subspecialties such as thoracic support and trauma nursing under the supervision of professional associations and certification bodies [32,33]. Featuring rigorous entry criteria and standardized curriculum structures, these frameworks have facilitated the progressive expansion of the scope of practice for specialist nurses in international contexts. Training and certification in China are primarily organized by the Chinese Nursing Association and provincial nursing associations. Despite over two decades of specialist nurse training experience, the development of thoracic surgery specialist nurses remains at an early stage. Existing studies, such as those by Zhang et al. [20] and Duan et al. [29], mainly address military training contexts, limiting generalizability. Notably, China first clarified the separation of cardiothoracic surgery into two independent specialties in 2008 [37]. Most previous research centered on cardiothoracic surgery specialist nurses, leaving the existing training system poorly aligned with the needs of specialist nurses in the newly established thoracic surgery specialty [22,23,25–28]. Although a handful of studies have explored thoracic surgery specialist nurses' training from specific perspectives [24,30], they generally lack systematic training systems, resulting in prominent issues including uneven multidisciplinary collaboration capacity, non-unified and non-standardized curriculum design, and insufficient content relevance [8]. To address these gaps, this study undertook the systematic development of a core competency training system for thoracic surgery specialist nurses, incorporating both national policy objectives and the unique clinical nuances of thoracic surgery. By establishing a structured framework encompassing recruitment, training, supervision, and evaluation, this system fosters the standardized advancement of thoracic surgical nursing practice, thereby aligning with national initiatives aimed at enhancing specialized medical service capabilities.

### Scientific validity and reliability of the core competencies training system

Guided by the well-established core competency theory and the Tyler Model, this study first systematically synthesized relevant domestic and international literature, policy documents, and educational guidelines. Considering the global development of thoracic surgery specialist nurse training and China's national context, high-quality evidence was synthesized to outline a preliminary competency framework and construct the training system's item pool. To incorporate diverse stakeholder perspectives, semi-structured interviews were subsequently conducted with practicing thoracic nurses and healthcare experts. These in-depth exchanges revealed diverse perspectives between frontline healthcare providers and senior medical experts, further enriching the training system's item pool. Ultimately, a two-round Delphi method was conducted with 19 experts from hospitals and universities across five provinces and municipalities. The panel included specialists in nursing management, education, and clinical care, all holding deputy senior or higher titles, bachelor's degrees

**Table 4. Consultation results of the core competencies training system for thoracic surgery specialist nurses (core competency framework/ training objectives, curriculum content, along with their corresponding teaching methods).**

| Core competency framework (Training objectives) & Curriculum content | Teaching method | Average | Standard deviation | Variable coefficient |
|---|---|---|---|---|
| **I Practical competency in thoracic surgical nursing** | | 5.00 | 0.00 | 0.00 |
| I-1 Proficiency in professional theoretical knowledge | | 5.00 | 0.00 | 0.00 |
| I-1–1 Regional thoracic anatomy and physiology | a | 4.95 | 0.23 | 0.05 |
| I-1–2 General diagnostic and therapeutic principles for thoracic diseases | a | 4.84 | 0.38 | 0.08 |
| I-1–3 Perioperative nursing protocols for thoracic surgical conditions | a | 5.00 | 0.00 | 0.00 |
| I-1–4 Specialized pharmacological knowledge and administration standards in thoracic surgery | a | 5.00 | 0.00 | 0.00 |
| I-1–5 Specialized knowledge of thoracic surgical examinations and result interpretation | a,e | 4.74 | 0.45 | 0.09 |
| I-2 Specialized technical and practical competency | | 5.00 | 0.00 | 0.00 |
| I-2–1 Indications and operation protocols for specialized thoracic surgical equipment (mechanical expectoration devices, respiratory trainers, respiratory humidification therapy devices, red light therapeutic apparatus, enteral nutrition pumps, etc.) | b | 4.95 | 0.23 | 0.05 |
| I-2–2 Perioperative complications management for common thoracic surgical conditions (pulmonary infection, atelectasis, hemorrhage, persistent pulmonary air leak, bronchopleural fistula, chylothorax, esophageal anastomotic fistula, esophagotracheal fistula, venous thromboembolism, arrhythmia, etc.) | b,d,e | 5.00 | 0.00 | 0.00 |
| I-2–3 Management and nursing of patients with pulmonary diseases (lung cancer, pulmonary nodules, pulmonary bullae, etc.) | b,e | 5.00 | 0.00 | 0.00 |
| I-2–4 Management and nursing of patients with esophageal diseases (esophageal cancer, esophageal leiomyoma, esophageal stromal tumor, etc.) | b,e | 5.00 | 0.00 | 0.00 |
| I-2–5 Management and nursing of patients with tracheal diseases (tracheal tumors, tracheal foreign bodies, etc.) | b,e | 5.00 | 0.00 | 0.00 |
| I-2–6 Management and nursing of patients with mediastinal diseases (thymoma, mediastinal emphysema, etc.) | b,e | 5.00 | 0.00 | 0.00 |
| I-2–7 Management and nursing of patients with chest wall diseases (pectus excavatum, pectus carinatum, chest wall tumors, etc.) | b,e | 4.95 | 0.23 | 0.05 |
| I-2–8 Management and nursing of patients with thoracic trauma (rib fractures, hemopneumothorax, pulmonary contusion, etc.) | b,e | 4.95 | 0.23 | 0.05 |
| I-2–9 Management and nursing of patients with other thoracic surgical conditions (hyperhidrosis, empyema, etc.) | b,e | 4.89 | 0.32 | 0.07 |
| I-2–10 Common surgical procedures for thoracic diseases and postoperative nursing key points | a,b | 5.00 | 0.00 | 0.00 |
| I-2–11 Current status and prospects of lung transplantation | a | 4.89 | 0.32 | 0.07 |
| I-2–12 Development of preoperative auxiliary localization techniques for pulmonary subsolid nodules and related nursing | a,b | 4.84 | 0.50 | 0.10 |
| I-2–13 Advances in interventional therapy for thoracic tumor patients and nursing practice | a,b | 4.89 | 0.32 | 0.07 |
| I-2–14 Operative coordination and postoperative nursing for thoracentesis | a,b | 4.95 | 0.23 | 0.05 |
| I-2–15 Operative coordination and nursing for bedside fiberoptic bronchoscopic suctioning | a,b | 4.84 | 0.38 | 0.08 |
| I-2–16 Development of electromagnetic navigation bronchoscopy technology and nursing implications | a | 4.68 | 0.58 | 0.12 |
| I-2–17 Current status and prospects of neoadjuvant immunotherapy for non-small cell lung cancer | a | 4.68 | 0.58 | 0.12 |
| I-2–18 Nursing of thoracic surgical catheters (chest closed drainage tubes, mediastinal drainage tubes, gastric tubes, nasoenteric tubes, jejunostomy tubes, central venous catheters, PICC, etc.) | a,b | 4.89 | 0.32 | 0.07 |
| I-2–19 Perioperative pain management for thoracic surgical conditions | a,e | 4.95 | 0.23 | 0.05 |
| I-2–20 Perioperative airway management and pulmonary rehabilitation strategies for thoracic diseases | a,d,e | 4.89 | 0.32 | 0.07 |
| I-2–21 Perioperative nutritional management for thoracic surgical | a,e | 4.89 | 0.32 | 0.07 |
| I-2–22 Perioperative blood glucose management for thoracic surgical conditions | a,e | 4.89 | 0.32 | 0.07 |
| I-2–23 Enhanced recovery after surgery (ERAS) in thoracic surgery from a nursing perspective | a,g | 4.95 | 0.23 | 0.05 |
| I-2–24 Daytime ward management processes and applications for thoracic surgical procedures | a,e | 4.89 | 0.32 | 0.07 |
| I-2–25 Nursing care for thoracic surgical patients with comorbid chronic diseases (hypertension, diabetes mellitus, COPD, etc.) | a,d,e | 4.95 | 0.23 | 0.05 |

*(Continued)*

| Core competency framework (Training objectives) & Curriculum content | Teaching method | Average | Standard deviation | Variable coefficient |
|---|---|---|---|---|
| I-2–26 Continuing nursing care or transitional nursing programs for thoracic surgical conditions | a | 4.68 | 0.48 | 0.10 |
| I-2–27 Development process and global status of specialized nursing clinics in thoracic surgery | a,g | 4.42 | 0.69 | 0.16 |
| I-2–28 Reception processes and nursing implementation in thoracic surgical nursing clinics | a | 4.47 | 0.77 | 0.17 |
| I-2–29 Palliative care for patients with advanced thoracic malignancies | a | 4.16 | 0.77 | 0.19 |
| I-2–30 Application of Traditional Chinese Medicine (TCM) nursing techniques in thoracic surgery care | a,b | 4.26 | 0.65 | 0.15 |
| I-3 Nursing assessment competency | | 5.00 | 0.00 | 0.00 |
| I-3–1 Commonly used perioperative assessment tools in thoracic surgery | a,f | 4.89 | 0.32 | 0.07 |
| I-3–2 Clinical assessment of thoracic diseases (including laboratory test indicators) and adjunctive examinations | a | 4.89 | 0.32 | 0.07 |
| I-4 Emergency and critical care competency | | 5.00 | 0.00 | 0.00 |
| I-4–1 Operation of resuscitation equipment (defibrillators, ventilators, bag-valve-mask devices, etc.) | b,c, | 5.00 | 0.00 | 0.00 |
| I-4–2 Precision fluid management for critically ill thoracic surgical patients | a,e | 4.95 | 0.23 | 0.05 |
| I-4–3 Recognition and emergency nursing for patients with thoracic massive hemorrhage | a,c | 5.00 | 0.00 | 0.00 |
| I-4–4 Blood transfusion procedures and management in thoracic surgery | a,c | 4.84 | 0.50 | 0.10 |
| I-4–5 Recognition and emergency nursing for pulmonary embolism patients | a,c | 5.00 | 0.00 | 0.00 |
| I-4–6 Recognition and emergency nursing for cerebral infarction patients | a,c | 4.89 | 0.46 | 0.09 |
| I-4–7 Recognition and emergency nursing for myasthenic crisis in thoracic surgical patients | a,c | 4.89 | 0.46 | 0.09 |
| I-4–8 Recognition and emergency nursing for postoperative diabetic ketoacidosis in thoracic surgical patients | a,c | 5.00 | 0.00 | 0.00 |
| I-4–9 Emergency management of thoracic drainage tube dislodgement | a,c | 4.95 | 0.23 | 0.05 |
| I-4–10 Recognition and management of common arrhythmias in thoracic surgical patients | a,f | 5.00 | 0.00 | 0.00 |
| I-4–11 Interpretation of the latest cardiopulmonary resuscitation (CPR) guidelines and practical training | a,b,f | 4.95 | 0.23 | 0.05 |
| I-4–12 Key points and nursing care for transferring critically ill thoracic surgical patients | a,c | 4.95 | 0.23 | 0.05 |
| I-4–13 Use and management of emergency medications, psychotropic/narcotic drugs, and high-alert medications | a | 4.95 | 0.23 | 0.05 |
| I-5 Critical thinking competency | | 5.00 | 0.00 | 0.00 |
| I-5–1 Application of evidence-based nursing in thoracic surgical care | a,g | 4.68 | 0.48 | 0.10 |
| I-5–2 Cultivation and application of critical thinking competency in thoracic nursing practice | a,d | 4.84 | 0.38 | 0.08 |
| II **Guidance and training competency** | | 4.95 | 0.23 | 0.05 |
| II-1 Health education competency | | 4.95 | 0.23 | 0.05 |
| II-1–1 Individualized health education and discharge guidance for thoracic surgical patients | a | 4.89 | 0.32 | 0.07 |
| II-1–2 Production and dissemination of popularization of science for thoracic surgery care | a | 4.74 | 0.45 | 0.09 |
| II-1–3 Self-management guidance for thoracic surgical patients | a | 4.79 | 0.42 | 0.09 |
| II-2 Teaching and training competency | | 4.84 | 0.38 | 0.08 |
| II-2–1 Integration of professional ethics in clinical nursing teaching design | a | 4.79 | 0.42 | 0.09 |
| II-2–2 Methods and techniques for clinical nursing teaching | a,d | 4.89 | 0.46 | 0.09 |
| II-2–3 Hierarchical management and practice in clinical nursing education | a | 4.95 | 0.23 | 0.05 |
| II-2–4 Curriculum design for thoracic surgical nursing | a | 4.58 | 0.61 | 0.13 |
| III **Communication and coordination competency** | | 4.89 | 0.32 | 0.07 |
| III-1 Communication competency | | 4.95 | 0.23 | 0.05 |
| III-1–1 Communication methods and verbal/nonverbal communication skills | a | 4.89 | 0.32 | 0.07 |
| III-1–2 Recognition and management of medical-nursing disputes | a | 4.89 | 0.32 | 0.07 |
| III-2 Coordination competency | | 4.89 | 0.32 | 0.07 |
| III-2–1 Interdepartmental and multidisciplinary team collaboration protocols and processes | a | 4.74 | 0.56 | 0.12 |

*(Continued)*

**Table 4.** (Continued)

| Core competency framework (Training objectives) & Curriculum content | Teaching method | Average | Standard deviation | Variable coefficient |
|---|---|---|---|---|
| IV **Innovative practice competency** | | 4.68 | 0.48 | 0.10 |
| IV-1 Research and innovation competency | | 4.68 | 0.48 | 0.10 |
| IV-1–1 Implementation and reflection on the latest consensus guidelines and group standards in thoracic surgery | a,g | 4.95 | 0.23 | 0.05 |
| IV-1–2 Literature retrieval strategies and hands-on practice | a,f | 4.89 | 0.32 | 0.07 |
| IV-1–3 Application of literature management software | a,f | 4.89 | 0.32 | 0.07 |
| IV-1–4 Topic selection and research design in nursing research | a,g | 4.89 | 0.32 | 0.07 |
| IV-1–5 Topic selection and writing of case nursing reports | a,g | 4.95 | 0.23 | 0.05 |
| IV-1–6 Formatting and standards for research proposal development | a | 4.79 | 0.54 | 0.11 |
| IV-1–7 Fundamentals of medical statistics and software applications | a | 4.42 | 0.61 | 0.14 |
| IV-1–8 Standardized writing and submission of nursing research papers | a | 4.84 | 0.38 | 0.08 |
| IV-2 Practice translation competency | | 4.79 | 0.42 | 0.09 |
| IV-2–1 Evidence-based nursing and clinical practice integration | a,g | 4.84 | 0.50 | 0.10 |
| IV-2–2 Application and translation of nursing patents | a,d | 4.42 | 0.69 | 0.16 |
| V **Management competency** | | 4.63 | 0.60 | 0.13 |
| V-1 Planning competency | | 4.58 | 0.77 | 0.17 |
| V-1–1 Development and implementation of thoracic surgical nursing plans | a | 4.95 | 0.23 | 0.05 |
| V-1–2 Whole-patient journey management for thoracic surgical conditions | a | 4.95 | 0.23 | 0.05 |
| V-2 Organizational competency | | 4.42 | 0.84 | 0.19 |
| V-2–1 Formats and execution of clinical nursing rounds in thoracic surgery | a,e | 4.84 | 0.50 | 0.10 |
| V-2–2 Complex case discussions under the multidisciplinary team (MDT) approach instructional approach | a,e | 4.84 | 0.50 | 0.10 |
| V-2–3 Nursing workforce allocation and management | a | 4.53 | 0.70 | 0.15 |
| V-2–4 Design and practice of emergency response drills | a,c | 4.84 | 0.50 | 0.10 |
| V-3 Leadership competency | | 4.32 | 0.75 | 0.17 |
| V-3–1 Clinical pathway process management in thoracic surgical nursing | a | 4.84 | 0.38 | 0.08 |
| V-3–2 Ward nursing management under the primary nursing model in thoracic surgery | a | 4.79 | 0.54 | 0.11 |
| V-3–3 Development planning for thoracic subspecialty nursing | a | 4.58 | 0.61 | 0.13 |
| V-3–4 Leadership in nursing practice for thoracic surgery | a | 4.42 | 0.61 | 0.14 |
| V-4 Control Competencies | | 4.42 | 0.84 | 0.19 |
| V-4–1 Establishment and monitoring of thoracic nursing quality indicators | a | 4.79 | 0.54 | 0.11 |
| V-4–2 Monitoring and management of nursing safety incidents | a | 4.79 | 0.54 | 0.11 |
| V-4–3 Nosocomial infection control and occupational risk protection | a | 4.84 | 0.50 | 0.10 |
| V-4–4 Three-level nursing rounds for critically ill thoracic surgical patients | a,e | 4.74 | 0.56 | 0.12 |
| VI **Professional ethics and judgment competency** | | 4.79 | 0.54 | 0.11 |
| VI-1 Ethical decision-making competency | | 4.74 | 0.56 | 0.12 |
| VI-1–1 Medical laws, regulations, and institutional policies | a | 4.79 | 0.54 | 0.11 |
| VI-1–2 Nursing ethics and moral standards | a | 4.79 | 0.54 | 0.11 |
| VI-2 Professional literacy | | 4.89 | 0.46 | 0.09 |
| VI-2–1 Clinical practice of humanistic care in thoracic surgery | a | 4.79 | 0.54 | 0.11 |
| VI-2–2 Occupational stress and self-regulation for thoracic specialist nurses | a | 4.68 | 0.58 | 0.12 |
| VI-2–3 Professional spirit and social responsibility for thoracic specialist nurses | a | 4.79 | 0.54 | 0.11 |
| VI-2–4 Professional development and career planning for thoracic specialist nurses | a | 4.79 | 0.54 | 0.11 |

**Teaching Method:** a = Lecture-based theoretical teaching; b = Skills-oriented training methodology; c = Scenario simulation pedagogy; d = Experiential sharing approach; e = Case-driven discussion framework; f = Workshop-based interactive learning; g = Literature-driven research methodology.

**Table 5. Consultation results of the core competencies training system for thoracic surgery specialist nurses (organizational management and evaluation methods).**

| Categories | | Items | Average | Standard deviation | Variable coefficient |
|---|---|---|---|---|---|
| Organizational management | I Training paradigm | I-1 Total course hours: 480 hours (45 minutes per hour) | 5.00 | 0.00 | 0.00 |
| | | I-2 Integrating theoretical knowledge and practical skills | 4.95 | 0.23 | 0.05 |
| | | I-3 Duration: 3 months, comprising 1 month of theoretical learning and 2 months of practical training | 5.00 | 0.00 | 0.00 |
| | | I-4 Full-time commitment without work responsibilities | 4.84 | 0.38 | 0.08 |
| | II Admission criteria for trainees | II-1 Holding a valid nursing practice qualification certificate | 4.95 | 0.23 | 0.05 |
| | | II-2 Bachelor's degree or higher education level | 4.95 | 0.23 | 0.05 |
| | | II-3 Professional title of Nurse Practitioner or higher | 4.95 | 0.23 | 0.05 |
| | | II-4 At least 6 years of clinical nursing experience | 4.95 | 0.23 | 0.05 |
| | | II-5 At least 3 years of nursing experience in thoracic surgery | 4.95 | 0.23 | 0.05 |
| | | II-6 At least 2 years of clinical nursing teaching experience | 4.79 | 0.42 | 0.09 |
| | | II-7 Basic English proficiency for academic learning | 4.58 | 0.51 | 0.11 |
| | | II-8 Demonstrated high ethical standards and professional morality | 5.00 | 0.00 | 0.00 |
| | III Faculty selection criteria | III-1 Bachelor's degree or higher (nurses); master's degree or higher (physicians) | 5.00 | 0.00 | 0.00 |
| | | III-2 Intermediate professional title or above (nurses); associate senior or higher professional title (physicians) | 4.89 | 0.32 | 0.07 |
| | | III-3 At least 10 years of specialized experience in related specialties | 4.84 | 0.50 | 0.10 |
| | | III-4 At leas 5 years of teaching experience in academic institutions or clinical training settings | 4.79 | 0.42 | 0.09 |
| | | III-5 Proven integrity, professional ethics, and dedication to education | 5.00 | 0.00 | 0.00 |
| Evaluation methods | I Formative evaluation | I-1 Class participation | 4.95 | 0.23 | 0.05 |
| | | I-2 Case analysis | 4.95 | 0.23 | 0.05 |
| | | I-3 Chinese/English professional literature reading presentation | 4.68 | 0.48 | 0.10 |
| | | I-4 Health education popularization video/manual development | 4.84 | 0.38 | 0.08 |
| | II Summative evaluation | II-1 Specialty theoretical assessment | 5.00 | 0.00 | 0.00 |
| | | II-2 Specialty skills assessment: Objective Structured Clinical Examination (OSCE) | 5.00 | 0.00 | 0.00 |
| | | II-3 Case nursing report | 5.00 | 0.00 | 0.00 |
| | | II-4 Research proposal | 4.95 | 0.23 | 0.05 |

or above, and over ten years of professional experience. Both Delphi rounds achieved a 100% response rate, reflecting strong expert engagement. The exceptionally high Cr of 0.895 indicated their profound clinical acumen and administrative expertise, ensuring their capacity to provide constructive input for framework refinement. Kendall's coefficient in the first and second rounds was 0.141 and 0.210, respectively (all $P < 0.05$). Furthermore, after two rounds of consultation, all CVs were below 0.25, demonstrating strong consensus on indicator selection. By integrating evidence from the literature, stakeholder insights, and Delphi consensus, this study developed a theoretically grounded and scientifically rigorous core competency training system for thoracic surgery specialist nurses, advancing specialized nursing education in thoracic surgery.

## Comprehensiveness and specialization of the core competencies training system

Training objectives serve as a critical compass in the development of any training system. This study identified the primary objective as enhancing the core competencies of thoracic surgery specialist nurses. By integrating the unique clinical features of thoracic surgery, such as the coexistence of acute and chronic diseases, the combination of surgery and neoadjuvant therapy, and the care of both pediatric and elderly patients [38], the research team developed a core competency framework aligned with the ICN Core Competency Standards to guide training system design. The resulting framework comprises six first-level core competencies and 17 second-level indicators. It comprehensively covers professional knowledge, technical skills, social roles, and personal attributes, thereby reflecting the professionalism of thoracic surgical nursing practice and providing strong clinical guidance value.

The curriculum content of training constitutes the main body of the training system construction. Based on the core competency framework, a structured curriculum was developed with 92 third-level indicators, consistent with both thoracic surgery's specialized characteristics and the requirements of high-quality nursing development. Clinical practice competency is the fundamental capability for thoracic surgery specialist nurses, underpinning their ability to provide exceptional care, which is an attribute associated with thoracic surgery's high specialization, complexity, and rigorous requirements for nurses' professional technical proficiency [39]. Amid rising thoracic surgery volumes and a growing population of elderly patients, having complex, robust, specialized theoretical knowledge and practical skills is a prerequisite for continuous disease monitoring and predictive nursing, which can effectively mitigate the serious consequences of sudden disease changes. Additionally, strengthening emergency and critical care capabilities enables nurses to rapidly respond to post-operative adverse events, providing safe and effective comprehensive nursing support throughout the entire course of care. As healthcare shifts toward patient-centered, outcome-oriented evaluation and the ERAS concept becomes widely adopted, specialist nurses' roles have evolved beyond sole clinical practitioners to encompass health managers and educators [40]. Shorter hospital stays heighten the need for post-discharge rehabilitation and patient education. Thus, thoracic surgery specialist nurses must possess strong guidance and training competencies to design personalized respiratory rehabilitation programs, develop evidence-based follow-up plans, promote recovery, and prevent complications [41,42]. They are also responsible for mentoring junior nurses, facilitating professional development, and promoting knowledge transfer. Furthermore, effective multidisciplinary collaboration is vital to thoracic surgery, involving coordination among surgical, ICU, and rehabilitation teams [43]. As the central link bridging these teams and patients, thoracic surgery specialist nurses' communication and coordination competencies directly influence diagnostic and treatment efficiency, as well as rehabilitation outcomes. Efficient alignment with the needs of different disciplines facilitates the implementation of early inpatient rehabilitation interventions, while effective doctor-patient communication reduces clinical misunderstandings and enhances patient treatment adherence. Meanwhile, thoracic surgical technology is advancing rapidly in minimally invasive and diversified directions, with the application of new surgical procedures and equipment presenting new challenges for nursing practice. Thoracic surgery specialist nurses must integrate research into practice to provide evidence-based specialized care and advance the discipline [44]. Additionally, thoracic surgical specialist nurses assume managerial roles in team coordination and resource optimization. Enhancing their managerial competencies equips them to apply management thinking in scientific clinical resource allocation, nursing process standardization, and staffing optimization, translating these skills into core competencies for routine practice and providing robust support for the standardization and homogenization of departmental nursing quality. Given the high-intensity and high-risk nature of thoracic surgery, specialist nurses require strong ethical awareness, empathy, and self-regulation to maintain patient safety and professional integrity. For teaching methods, while lecture-based theoretical teaching remained the primary instructional modality, problem-oriented modules involving scenario simulation pedagogy, case-driven discussion framework, and workshop-based interactive learning were integrated for intervention development and protocol design. This blended approach created interactive learning environments that facilitated deep engagement and applied competency building, addressing the dual needs for knowledge acquisition and skill integration in specialized nursing training.

Training organizational management serves as a critical component in ensuring the effectiveness of specialist nurse training. To ensure program effectiveness, the training system specified clear criteria for the training paradigm, trainee admission, and faculty selection. A 1:2 ratio of theoretical to clinical training was adopted to enhance the integration of knowledge and practice. Rigorous admission standards were implemented to ensure a homogeneous learning foundation, encompassing professional qualifications and clinical experience. A bachelor's degree or higher was required, reflecting China's educational standards. Faculty selection emphasized clinical expertise, pedagogical competence, and academic rank to ensure instructional quality and establish a replicable model for future research.

Objective and rational assessment is a critical benchmark for evaluating the effectiveness of training in specialist nurse education. A comprehensive evaluation framework was established to objectively assess training outcomes. Combining formative and summative assessments, the system measured clinical proficiency, research capability, and managerial competence. Formative evaluation emphasized ongoing development through classroom participation, case analysis, and literature reviews. Summative evaluation included theoretical and practical examinations, case reports, and research proposals. The integration of both subjective and objective indicators ensured fairness, consistency, and reliability in performance evaluation, providing a scientific reference for competency-based assessment in thoracic surgical nursing.

## Impact

From a nursing management perspective, this training system functions as a standardized framework for the enrollment, training, and assessment of thoracic surgery specialist nurses, providing core support for national qualification certification in this field. It further promotes the standardized development of thoracic surgical nursing as a distinct discipline and lays a solid foundation for building high-quality professional talent teams. For practicing nurses, the system facilitates the identification of strengths and improvement areas, enabling targeted enhancement of their core professional competencies. In terms of patient care, its implementation effectively optimizes nursing quality, thereby improving the medical experience of thoracic surgery patients and accelerating their rehabilitation. Regarding the healthcare system, the training system boasts robust scalability and promotion prospects. Against this backdrop, the Jiangsu Provincial Nursing Association launched the inaugural thoracic surgery specialist nurse training program in 2024. As an officially accredited training base, our hospital took the lead in applying the self-developed training system in August 2025 to verify its clinical applicability. Qualification certificates will be awarded to participants who successfully complete the training and assessment. Notably, this system can subsequently be promoted to national thoracic surgery specialist nurse training programs and is expected to serve as a replicable template for other nursing specialties, fostering the high-quality development of China's nursing industry. In recent years, with the continuous expansion of China's specialized nursing workforce, several medical institutions and nursing colleges have actively explored the Advanced Practice Nurse (APN) training model. Adopting a core admission criterion of a master's degree plus specialist nurse certification, this model aims to enhance theoretical and practical capabilities while constructing multi-level career development pathways. Therefore, this training system holds significant practical value for exploring thoracic surgery APN and aligning with international standards.

## Limitations

Firstly, although the Delphi experts were recruited from fields such as thoracic surgery nursing, medicine, education, and management across five provinces or municipalities, broader representation from more fields and regions would enhance validity. Secondly, due to time constraints, the constructed system remains at the theoretical level. Future research should focus on long-term impacts and broader application scenarios, employing large-scale empirical research to validate its practical value and further optimize the system's effectiveness. Additionally, given the varying levels of development of specialist nursing across countries, the international applicability of this system requires further validation.

## Conclusion

This study established a relatively scientific, comprehensive, and specialty-specific training system for thoracic surgery specialist nurses. It both addresses the unique challenges in China's clinical context and aligns closely with the practical job requirements of thoracic surgical nursing, serving as a valuable addition to the emerging evidence base in this field and boasting potential clinical significance and practical application value. Future research could further link up with academic institutions to improve the early training mechanism for thoracic surgery specialist nurses. Meanwhile, empirical research should be adopted to strengthen the continuous feedback and iteration of the training system, enabling it to dynamically adapt to changes in domestic and international healthcare needs and further expand its practical influence and application value.

## Supporting information

**S1 File.**
(ZIP)

## Acknowledgments

We extend our gratitude to the 19 experts and all the participants who volunteered to participate in the study during their busy schedules.

## Author contributions

**Conceptualization:** Yingjin Li, Yan Chen, Qiuju Chen.

**Data curation:** Yingjin Li, Di Zhu.

**Formal analysis:** Yingjin Li, Di Zhu.

**Investigation:** Qiuju Chen, Ping Xia, Yuqin Mao, Liumei Yin, Saijun Ba.

**Methodology:** Di Zhu.

**Project administration:** Yan Chen, Qiuju Chen.

**Resources:** Yan Chen.

**Supervision:** Yan Chen.

**Visualization:** Yuqin Mao, Liumei Yin, Saijun Ba.

**Writing – original draft:** Yingjin Li, Di Zhu.

**Writing – review & editing:** Yan Chen, Qiuju Chen, Ping Xia.

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
