## [Decision Letter · Decision Letter 0]

24 Oct 2025

Dear Dr. Li,

Please note that we have only been able to secure a single reviewer to assess your manuscript. We are issuing a decision on your manuscript at this point to prevent further delays in the evaluation of your manuscript. Please be aware that the editor who handles your revised manuscript might find it necessary to invite additional reviewers to assess this work once the revised manuscript is submitted. However, we will aim to proceed on the basis of this single review if possible.Please review the comments provided below and make the appropriate revisions to address all concerns raised. This includes a more thorough discussion of the literature and an expanded limitations and conclusions section among other recommendations.

We look forward to receiving your revised manuscript.

Kind regards,

Emma Campbell, Ph.D

Staff Editor

PLOS ONE

**Journal Requirements:**

1. When submitting your revision, we need you to address these additional requirements. Please ensure that your manuscript meets PLOS ONE's style requirements, including those for file naming. The PLOS ONE style templates can be found at https://journals.plos.org/plosone/s/file?id=wjVg/PLOSOne_formatting_sample_main_body.pdf and https://journals.plos.org/plosone/s/file?id=ba62/PLOSOne_formatting_sample_title_authors_affiliations.pdf 2. We note that you have indicated that there are restrictions to data sharing for this study. For studies involving human research participant data or other sensitive data, we encourage authors to share de-identified or anonymized data. However, when data cannot be publicly shared for ethical reasons, we allow authors to make their data sets available upon request. For information on unacceptable data access restrictions, please see http://journals.plos.org/plosone/s/data-availability#loc-unacceptable-data-access-restrictions.  Before we proceed with your manuscript, please address the following prompts: a) If there are ethical or legal restrictions on sharing a de-identified data set, please explain them in detail (e.g., data contain potentially identifying or sensitive patient information, data are owned by a third-party organization, etc.) and who has imposed them (e.g., a Research Ethics Committee or Institutional Review Board, etc.). Please also provide contact information for a data access committee, ethics committee, or other institutional body to which data requests may be sent. b) If there are no restrictions, please upload the minimal anonymized data set necessary to replicate your study findings to a stable, public repository and provide us with the relevant URLs, DOIs, or accession numbers. Please see http://www.bmj.com/content/340/bmj.c181.long for guidelines on how to de-identify and prepare clinical data for publication. For a list of recommended repositories, please see https://journals.plos.org/plosone/s/recommended-repositories. You also have the option of uploading the data as Supporting Information files, but we would recommend depositing data directly to a data repository if possible. Please update your Data Availability statement in the submission form accordingly. 3. If the reviewer comments include a recommendation to cite specific previously published works, please review and evaluate these publications to determine whether they are relevant and should be cited. There is no requirement to cite these works unless the editor has indicated otherwise. 

Reviewers' comments:

**Comments to the Author**

1. Is the manuscript technically sound, and do the data support the conclusions?

Reviewer #1: Partly

2. Has the statistical analysis been performed appropriately and rigorously?

Reviewer #1: Yes

3. Have the authors made all data underlying the findings in their manuscript fully available?

Reviewer #1: No

4. Is the manuscript presented in an intelligible fashion and written in standard English?

Reviewer #1: Yes

**Reviewer #1:**  The authors have chosen a highly relevant topic and highlighted it with a complex scientific approach which seems to be an indicator of how seriously the authors set themselves apart with the field of nursing specialisation within thoracic surgery.

Sound condensed abstract. However, the conclusion section should not only focus on the scientific rigor. It should highlight clinical implications and (as already done) recommendations for future (research) steps. Also make sure to not contradict yourself (compare with limitations section).

Titles: Should follow the same structure through the whole study. Please check for standardised capitalisation throughout the whole paper.

During the introduction a well written rational can be followed which is identifying the research gap and problem and finally leading to the research problem. Well done!

The methodology section is overall very well written. However, for the literature review it is lacking at least the final search protocols for each database (the authors could also place this in the appendix). Also regarding the literature review, the authors should be aware that including only free full texts is a high risk for selection bias. This should be addressed within the limitations section. Moreover, the whole study is lacking a discussion of the identified literature. The studies were not included within the appendix nor were they part of the results section or at least clearly flagged as such. For clarity the identified 14 studies should be appraised and critically discussed as usually done within literature reviews. Also verify how the literature was appraised (e.g. using a validated quality assessment tool). The methodologies are frequently repeated within the results and discussions. This seems unnecessary. Overall, the methodology section very detailed and as already mentioned, it is recommended to present the literature review findings within the results section and then critically discuss these findings within the discussion section.

It remains unclear on what level the core curriculum is set (e.g. Master, Diploma level). To comply with international standards for such nursing specialists the International Council of Nurses recommends a degree at master’s level. The authors should specify and justify (e.g. local context) how the role was located. Moreover, it remains unclear to the reader if the role of the thoracic surgical nurse specialist is an advanced practice role or a clinical specialisation on basic nursing level. For international comparability this should be declared clearly. Furthermore, the description as thoracic surgical nurse specialist might be misleading, as surgical nurse often refers to a scrub nurse or a nurse which is part of the intraoperative surgical team. Therefore, nomenclature should be reconsidered (if possible). If on advanced practice level, a more widespread naming is advanced practice nurse / nurse practitioner in thoracic surgery. This would add knowledge to the international advanced practice nursing community with a clear understanding of the here discussed role.

The limitations and conclusion sections remain very superficial. The authors should not only describe what or how they did something but interpret their findings, what the clinical implications are and identify future research focus.

Recommendations for reworking specific sections

Line 45 please provide reference

Line 51 newer evidence on complication rates are available for thoracic surgery. Consider using/searching this evidence.

Line 82 complete reference

Line 193 please provide reference for Colaizzi

Line 372 please provide reference

Line 373 please provide reference

Line 397 complete reference

Overall, the subheadings within the methodology section and the results section are quite complex which is at some stage reader unfriendly and sometime do not help the reader to understand what will be discussed in the sub chapter. Simplify these titles could support reader friendliness.

The authors of the study identified a valuable topic with international interest. They applied a well thought scientific approach. From the reviewer’s perspective, this study is considered suitable for publication. It can add to the emerging evidence in enhancing specialised nursing training in thoracic surgery and specific challenges in the Chinese context. However, the reviewer is recommending to address the above-mentioned feedback before publication.

**Do you want your identity to be public for this peer review?** For information about this choice, including consent withdrawal, please see our Privacy Policy

Reviewer #1: No

---

## [Author Response · Author response to Decision Letter 1]

12 Nov 2025

Dear Editor and Reviewer,

We are deeply grateful for the time and effort you dedicated to providing the insightful feedback and suggestions for improving our manuscript titled “Construction of the core competencies training system for thoracic surgery specialist nurses: A mixed-methods study” (Manuscript Number: PONE-D-25-47557). These comments are invaluable, and they have been immensely instrumental in enhancing the quality of this nursing-focused manuscript and providing critical guidance for our research. We have addressed each comment with a detailed, evidence-based response, and all revisions have been highlighted in yellow within the submitted revised documents for clear identification. We have made every effort to ensure the clarity and transparency of all revisions, and we hope the revised manuscript fully meets the journal’s publication criteria.

Response to Editor:

1. When submitting your revision, we need you to address these additional requirements. Please ensure that your manuscript meets PLOS ONE’s style requirements, including those for file naming.

Response: We sincerely appreciate your meticulous proofreading. We apologize for the formatting inconsistencies and nomenclature errors present in the original manuscript. Guided by the PLOS ONE Style Templates and nursing academic writing standards, we have thoroughly reviewed the manuscript repeatedly and corrected all formatting and naming inconsistencies.

2.We note that you have indicated that there are restrictions to data sharing for this study. For studies involving human research participant data or other sensitive data, we encourage authors to share de-identified or anonymized data. However, when data cannot be publicly shared for ethical reasons, we allow authors to make their data sets available upon request. Please update your Data Availability statement in the submission form accordingly.

Response: Thank you for your kind suggestion. All data from our study are available for sharing in full compliance with ethical guidelines for human subjects research. We have revised the “Data availability statement” (Page 30, Lines 500-502) as requested. The corresponding data have been uploaded as a Supplementary File (Page 36; Lines 631-638) to the manuscript submission system, with appropriate citations and references inserted at the relevant positions in the main text. If you consider that any additional data are required to support the manuscript, please do not hesitate to contact us. We will actively cooperate and make every effort to provide all relevant original data while adhering to nursing research ethical standards and data protection regulations.

3.If the reviewer comments include a recommendation to cite specific previously published works, please review and evaluate these publications to determine whether they are relevant and should be cited. There is no requirement to cite these works unless the editor has indicated otherwise.

Response: We sincerely appreciate your reminder regarding the citation guidelines. Upon a thorough review of the reviewers’ comments, no specific recommendations for citing previously published literature relevant to this nursing research were identified.

Response to Reviewer:

First and foremost, we would like to express our sincere gratitude for your positive feedback on this manuscript and valuable constructive suggestions. The responses to your corresponding suggestions are presented below, and we sincerely request your review and guidance.

1.Sound condensed abstract. However, the conclusion section should not only focus on the scientific rigor. It should highlight clinical implications and (as already done) recommendations for future (research) steps. Also make sure to not contradict yourself (compare with limitations section).

The limitations and conclusion sections remain very superficial. The authors should not only describe what or how they did something but interpret their findings, what the clinical implications are and identify future research focus.

Response: We are grateful for your valuable constructive comments. We fully agree that the original manuscript had a notable gap in elaborating on clinical implications and future research directions. To address this comment, we have added a dedicated “Impact” section (Page 27; Page 28, Lines 453-454) to systematically emphasize the clinical implications of our findings from four core nursing-related perspectives: nursing administrators, practicing nurses, patients, and the healthcare system. Furthermore, combined with the current status of the Thoracic surgery specialist nurse training program in Jiangsu Province, we have clarified the practical application prospects and specific future research directions of this study. In response to the consistency concern, we have thoroughly reviewed the conclusion and limitations sections. We have revised the limitations section to eliminate any potential inconsistencies, ensuring that the conclusions are logically aligned with the acknowledged study constraints (Page 28, Lines 456-464). Finally, we have succinctly summarized the above supplementary content in the conclusion to enhance its comprehensiveness and relevance (Page 2, Lines 30-36; Page 28, Lines 466-474; Page 29, Line 475).

2.Titles: Should follow the same structure through the whole study. Please check for standardised capitalisation throughout the whole paper.

Response: Thank you so much for your careful check. We apologize for the formatting inconsistencies present in the original manuscript. To address this comment comprehensively, we have conducted a thorough and repeated review of the entire manuscript, rectifying all identified formatting issues, including but not limited to standardizing the structure of all titles and ensuring consistent capitalization across all sections, in full compliance with the PLOS ONE Style Templates and nursing academic writing norms.

3.The methodology section is overall very well written. However, for the literature review it is lacking at least the final search protocols for each database (the authors could also place this in the appendix). Also regarding the literature review, the authors should be aware that including only free full texts is a high risk for selection bias. This should be addressed within the limitations section. Moreover, the whole study is lacking a discussion of the identified literature. The studies were not included within the appendix nor were they part of the results section or at least clearly flagged as such. For clarity the identified 14 studies should be appraised and critically discussed as usually done within literature reviews. Also verify how the literature was appraised (e.g. using a validated quality assessment tool). The methodologies are frequently repeated within the results and discussions. This seems unnecessary. Overall, the methodology section very detailed and as already mentioned, it is recommended to present the literature review findings within the results section and then critically discuss these findings within the discussion section.

Response: We sincerely appreciate your positive feedback on the methodology section of the manuscript and your detailed, professional suggestions for revision. We have addressed each of your comments comprehensively as follows:

Regarding the literature review-related comments:

(a)To address the missing search protocols, we have supplemented detailed literature search strategies, including search retrieval formulas, and initial retrieval results for each database in S1 File for transparent reference.

(b)Notably, we only excluded studies for which full texts were unavailable due to outdated publication dates, rather than restricting inclusion to free full texts. Given the potential misunderstanding arising from our previous expression, we revised Table 2 to clarify the specific exclusion criteria for full-text screening.

(c)Concerning literature quality appraisal: We used a validated quality assessment tool to conduct a rigorous evaluation of the included studies. Detailed descriptions of the tool and appraisal process are provided in the manuscript (Page 6, Lines 121-124; Page 7, Lines 125-127), with specific appraisal results for each included study documented in S2 File.

(d) Regarding the presentation and discussion of literature findings: We have integrated the literature review results into the Results section, with proper citations to all 14 included studies (Page 11, Lines 204-207). Furthermore, we have added a critical discussion of these studies in the Discussion section (Page 21, Lines 310-320; Page 22, Lines 321-326), which further supports the purpose of this study and the practical significance of constructing the core competencies training system for thoracic surgery specialist nurses.

Regarding redundant methodological descriptions: We have systematically reviewed the Methods, Results, and Discussion, removing redundant and repetitive methodological details to enhance the manuscript’s conciseness and readability.

4.It remains unclear on what level the core curriculum is set (e.g. Master, Diploma level). To comply with international standards for such nursing specialists the International Council of Nurses recommends a degree at master’s level. The authors should specify and justify (e.g. local context) how the role was located. Moreover, it remains unclear to the reader if the role of the thoracic surgical nurse specialist is an advanced practice role or a clinical specialisation on basic nursing level. For international comparability this should be declared clearly. Furthermore, the description as thoracic surgical nurse specialist might be misleading, as surgical nurse often refers to a scrub nurse or a nurse which is part of the intraoperative surgical team. Therefore, nomenclature should be reconsidered (if possible). If on advanced practice level, a more widespread naming is advanced practice nurse/nurse practitioner in thoracic surgery. This would add knowledge to the international advanced practice nursing community with a clear understanding of the here discussed role.

Response: We sincerely appreciate your critical and insightful suggestions, which have helped us refine the clarity and international relevance of the role definition. Consistent with the development status of nursing in China, a “specialist nurse” is defined as a registered nurse who possesses solid theoretical knowledge and proficient clinical skills in a specific nursing specialty, has completed and passed systematic specialist nurse training programs, and is capable of delivering high-quality specialized nursing care. Accordingly, the role of “thoracic surgery specialist nurse” in this study is positioned as a clinical specialisation on basic nursing level, with the core curriculum designed for nurses holding a bachelor’s degree or above. To clarify this positioning, we have supplemented detailed explanations in the Introduction (Page 3, Lines 56-58; Page 4, Lines 59-62) and further elaborated on its rationality in combination with the training system framework in the Discussion section (Page 26, Lines 417-420), highlighting alignment with China’s current nursing education and clinical practice context. Furthermore, In China’s healthcare system, intraoperative nursing care for thoracic surgery is exclusively provided by dedicated operating room nurses (scrub nurses or circulating nurses). The “thoracic surgery specialist nurse” in this study focuses on perioperative nursing management, disease-specific health education, complication prevention, and long-term follow-up care for thoracic surgery patients (Page 4, Lines 67-69). To avoid international misunderstandings and align with the role’s core responsibilities, we have revised the terminology to “thoracic surgery specialist nurse” throughout the manuscript.

5.Recommendations for reworking specific sections:

Line 45 please provide reference.

Line 51 newer evidence on complication rates are available for thoracic surgery. Consider using/searching this evidence.

Line 82 complete reference.

Line 193 please provide reference for Colaizzi.

Line 372 please provide reference.

Line 373 please provide reference.

Line 397 complete reference

Response: We highly appreciate your meticulous suggestions on this manuscript. We sincerely apologize for the errors in some of the references. We have uniformly revised and improved both the citation and formatting of the references in the manuscript, and conducted a final check prior to submission via the online submission system to ensure their proper display. Additionally, in response to your suggestion regarding postoperative complication rates in thoracic surgery, we have conducted a supplementary search for the latest high-quality evidence. The corresponding content in the manuscript has been updated and enhanced with these new findings (Page 3, Lines 50-55), ensuring the manuscript’s timeliness and evidential rigor.

6.Overall, the subheadings within the methodology section and the results section are quite complex which is at some stage reader unfriendly and sometime do not help the reader to understand what will be discussed in the sub chapter. Simplify these titles could support reader friendliness.

Response: We highly appreciate your valuable feedback regarding the readability of subheadings in the Methodology and Results sections. We have systematically reviewed and streamlined all subheadings. Specifically, we have simplified the terminology, removed redundant descriptive phrases, and restructured the titles to be concise, direct, and logically reflective of the key content within each subsection, ensuring readers can easily understand what is discussed in the subchapter.

Thank you again for giving us the opportunity to strengthen our manuscript with your valuable comments and queries. We deeply appreciate all your the valuable comments and suggestions, and look forward to hearing from you regarding our submission. We would be glad to respond to any further questions and comments that you may have.

Best regards.

Yours sincerely,

Yingjin Li

---

## [Decision Letter · Decision Letter 1]

11 Dec 2025

Construction of the core competencies training system for thoracic surgery specialist nurses: A mixed-methods study

PONE-D-25-47557R1

Dear Dr.Yingjin Li

We’re pleased to inform you that your manuscript has been judged scientifically suitable for publication and will be formally accepted for publication once it meets all outstanding technical requirements.

Kind regards,

Vinit Kumar Ramawat, M.Sc Nursing

Guest Editor

PLOS One

Additional Editor Comments (optional):

Reviewers' comments:

Reviewer's Responses to Questions

**Comments to the Author**

Reviewer #1: All comments have been addressed

2. Is the manuscript technically sound, and do the data support the conclusions?

Reviewer #1: Yes

3. Has the statistical analysis been performed appropriately and rigorously?

Reviewer #1: N/A

4. Have the authors made all data underlying the findings in their manuscript fully available?

Reviewer #1: Yes

5. Is the manuscript presented in an intelligible fashion and written in standard English?

Reviewer #1: Yes

Reviewer #1: (No Response)

**Do you want your identity to be public for this peer review?** For information about this choice, including consent withdrawal, please see our Privacy Policy

Reviewer #1: No

---

## [Editor Report · Acceptance letter]

PONE-D-25-47557R1

PLOS One

Dear Dr. Chen,

I'm pleased to inform you that your manuscript has been deemed suitable for publication in PLOS One. Congratulations! Your manuscript is now being handed over to our production team.

Kind regards,

on behalf of

Prof. Vinit Kumar Ramawat

Guest Editor

PLOS One